# Plasmapheresis for a Patient with Catatonia and Systemic Lupus Erythematosus: A Case Report and Literature Review

**DOI:** 10.3390/jcm11226670

**Published:** 2022-11-10

**Authors:** Pei-Shan Tsai, Yu Chen, Shou-Yen Chen, Chung-Yuan Hsu, Jiao-En Wu, Chih-Chun Lee, Tien-Ming Chan

**Affiliations:** 1Taipei Tzu Chi Hospital, Division of Gastroenterology, New Taipei City 231, Taiwan; 2Division of Rheumatology, Allergy and Immunology, New Taipei Municipal Tucheng Hospital, Chang Gung Memorial Hospital, New Taipei City 236, Taiwan; 3Department of Emergency Medicine, Chang Gung Memorial Hospital, Linkou Branch, Taoyuan 333, Taiwan; 4Division of Rheumatology, Allergy and Immunology, Chang Gung Memorial Hospital, Kaohsiung City 833, Taiwan; 5Division of Hematology-Oncology, Chang Gung Memorial Hospital, Linkou Branch, Taoyuan 333, Taiwan; 6Department of Medical Education, Chang Gung Memorial Hospital, Keelung Branch, Keelung City 204, Taiwan; 7Division of Rheumatology, Allergy and Immunology, Chang Gung Memorial Hospital, Linkou Branch, Chang Gung University, Taoyuan City 333, Taiwan

**Keywords:** SLE, catatonia, plasmapheresis

## Abstract

Neuropsychiatric systemic lupus erythematous (NPSLE) encompasses various psychiatric and neurological manifestations that develop in patients with systemic lupus erythematous (SLE), secondary to the involvement of the central nervous system (CNS). Although neuropsychiatric manifestations are commonly described in NPSLE, catatonia has been less frequently reported in patients with SLE. The roles of benzodiazepines (BZDs), immunosuppression, therapeutic plasma exchange (TPE), and electroconvulsive therapy (ECT) have all been reported in the management of catatonia. Furthermore, another research reported that catatonic symptoms associated with NPSLE were considerably improved by TPE. We, herein, report a case of catatonia in a patient with newly diagnosed NPSLE who exhibited a favorable prognosis through the early initiation of systemic immunosuppressants and TPE. Furthermore, we have reviewed the literature on the role of medication and plasmapheresis (PP), or TPE, in the treatment of catatonia that is associated with SLE.

## 1. Case Presentation

An 18-year-old female patient was referred to the emergency department (ED) of Chang Gung Memorial Hospital, Linkou branch, for acute psychotic symptoms, associated yelling, and refusal of oral intake for 2 days. The patient had been experiencing amenorrhea for 2 months. Eager to relieve the symptoms, she had taken menstruation-inducing agents. She claimed that she had not had any recent sexual activity. She had exhibited a puffy face for the 2 weeks before presentation. Associated symptoms included hair loss and discolouration of her palms. Owing to the progressive symptoms, she had visited a local hospital. In laboratory examinations at local hospital, positive findings for antinucleotide antibody (ANA) (1:320), anti-double-strand DNA (433 IU/mL. normal range <10 IU/mL), and low complement C3 (36 mg/dL. Reference: 90–180 mg/dL)/C4 (2.9 mg/dL. Reference: 10–40 mg/dL) levels were noted. According to medical records by local hospital, proteinuria, which is 3+ by urine stick examination, was noted as well. With an initial diagnosis of SLE, oral steroid therapy was first prescribed by local hospital.

After the visit at local hospital with the diagnosis of systemic lupus erythematous (SLE), her parents took her to our rheumatology outpatient department (OPD) for a second opinion. A retrospective evaluation of medical history and laboratory data re-confirmed the diagnosis of SLE by EULAR/ACC criteria (2019). Oral prednisolone (40 mg/day) was prescribed for the impression of lupus activity with OPD tracing a week later. At night, the patient experienced insomnia and depressive mood. A day after presentation, she started experiencing nervousness and the progression of several psychotic symptoms. She began speaking incoherently and loudly to herself, and she experienced persecutory delusions. Her mood changed rapidly and without any warning signs. She started yelling hysterically and refused any oral intake. Negative thinking was expressed without a suicide attempt. That night (of the day after presentation), she exhibited several twitching movements regularly in her legs and irregularly in her hands. Symptoms lasted for seconds at a time and did not involve loss of consciousness, according to her parents. Owing to prolonged and progressive symptoms, she was returned to the rheumatology OPD within 48 h and was later transferred to the ED for acute psychotic symptoms.

On physical examination, her consciousness was assessed as disorientation to time and place, with a Glasgow coma scale score of E4V4M6. A neurological examination revealed largely typical results; however, a complete evaluation was not possible due to the patient’s agitated status. Computed tomography and magnetic resonance angiography of the brain suggested vasculitis in bilateral high frontal subcortical white matter. Laboratory examinations revealed mild normocytic anaemia with haemoglobin concentration and haematocrits levels of 9.7 g/dL and 28.8%, respectively. Creatinine level was 0.54 mg/dL. There were no evidence with active infection by urine, blood and cerebrospinal fluid studies. Morphine, ketamine, methylenedioxymethamphetamine (MDMA), and amphetamine tests were negative. Repeated urine analysis revealed no proteinuria. However, ANA (speckled; 1:1280), anti-dsDNA (362.7 IU/mL), and ribosome-P (178.45 units) were positive. Besides low values of complements in C3 (40 mg/dL) and C4 (1 mg/dL), and high anti-phospholipid IgG (29.75 GPL. Reference: <15 GPL) were revealed. The hospital’s psychiatric department was urgently consulted after initial diagnosis of lupus encephalitis in the ED. Empirical pulse therapy was administered with methylprednisolone 1000 mg once per day for 3 days, with improving condition but still a poor response to stimulation.

After admission to the rheumatology ward, a nephrologist was consulted for plasmapheresis (PP), which is supplied by Taiwan national health insurance (TW-NHI) system, with the diagnosis of lupus with central nerve system involvement. PP was completed 5 times—on days 4, 6, 8, 10 and 12—after admission. Immunosuppressant therapy with hydrocortisone, azathioprine, and hydroxychloroquine (HCQ) were adjusted by clinical condition. Additionally, with symptoms of improved consciousness after lorazepam or midazolam infusion at ED, the neurologist was consulted to rule out possible epilepsy events. A cerebrospinal fluid study demonstrated negative findings for infection. Electroencephalography performed 3 times demonstrated negative findings. Consciousness returned to E4V4M6 (Glasgow coma scale score) after the fourth PP.

Moreover, progressive consciousness disturbance was noted. Antibiotic treatment was maintained to prevent infection. Gradually, fever conditions abated over the duration of antibiotic therapy. In the laboratory examinations, conditions of leukocytosis and elevated C-reactive protein levels had been alleviated by the tracing on day 17. However, the patient’s unstable mood, slow response, and incoherent speech remained despite improving objective examinations. Hence, the psychiatrist was consulted again on day 19. The patient demonstrated occasional stupor, posturing, and excitement during the psychiatric interview. Inappropriate emotional fluctuations, waxy flexibility, and negativism were noted. With serial symptoms and positive result by lorazepam test, catatonia was diagnosed. Due to CNS lupus complicated by catatonia, PP was recommended by the rheumatologist. Regular oral lorazepam (2 mg, 4 times/day) was recommended for symptom control. Impressively, the patient’s consciousness recovered to normal (E4V5M6) within 1 week of commencing PP therapy (Figure 1). Lorazepam therapy is omitted with improving symptoms. There are no further recurrent psychotic symptoms. In a stable state, the patient was discharged on day 28 with outpatient follow-up with a rheumatological profile followed and oral steroid (Figure 2). Further medication with cyclophosphamide is adjusted in outpatient clinic.

## 2. Discussion

In this report, we present a case of newly diagnosed SLE (CNS lupus) comorbid with catatonia. Furthermore, the symptoms were well controlled by immunosuppressive therapy followed by PP. The diagnosis of catatonia, which has been less frequently found in SLE [1], was made after performing a comprehensive multidisciplinary assessment, which included psychiatric, neurological, rheumatological, and nephrological evaluations.

Due to underlying SLE, the patient was initially diagnosed and treated as having lupus with CNS involvement. However, with persistent symptoms of fluctuant consciousness, and other newly developed symptoms, the patient was eventually diagnosed as having catatonia through a lorazepam test and was successfully treated with immunosuppressive therapy followed by PP and BZDs. This case highlights the need for vigilant awareness, recognition, and management of this syndrome.

### 2.1. Early Diagnosis of Catatonia Was the Best Policy

Neuropsychiatric NPSLE is characterised by 19 heterogeneous central and peripheral nervous system manifestations. It generally occurs in 30% to 40% of patients with SLE [2], and 50% to 60% of cases occur within the first year of a formal SLE diagnosis [3]. Specific manifestations, such as cognitive dysfunction, and psychiatric and movement disorders are uncommon, with recorded incidences of 1% to 5% and less than 1%, respectively [3]. A review article by Boeke et al., identified 37 SLE cases in which patients presented with psychomotor and behavioural abnormalities over a span of 4 decades, suggesting that catatonia is a rare neuropsychiatric disorder in patients with SLE [1]. However, catatonia has not been classified as a clinical feature of NPSLE. Catatonia improves with sedative anticonvulsants (barbiturates and BZDs) and, in severe cases, with ECT [4]. Catatonic symptoms associated with NPSLE have been reported to considerably improve with TPE [5].

A useful diagnostic test involves the intravenous administration of 1 to 2 mg of lorazepam, with a positive finding of dramatic recovery of spontaneous speech and movement disorders within 5 to 10 min [6]. In addition, high-dose BZDs or ECT are recommended for non-malignant or rapidly progressing catatonic symptoms [7]. In our case, catatonia was initially controlled in this manner for sedation before arranged examination.

Catatonia is a neuropsychiatric syndrome accompanied by psychomotor symptoms that result in abnormal movements and behaviours [8,9]. Currently, the diagnostic approach relies on the Diagnostic and Statistical Manual of Mental Disorders, Fifth Edition (DSM-5), which specifies that a diagnosis of catatonia be made for patients exhibiting 3 out of 10 specific symptoms and more than 3 of 12 features (Table 1) [8]. Secondary catatonia may be related to general medical conditions, certain psychotic diseases, mood disorders, and other unspecified causes [10]. In the DSM-5, catatonia is considered to arise from medical, psychiatric, or unspecified sources [8,11]. Catatonia is categorized into three subgroups by its symptoms [8,9]. All three subtypes of catatonia are defined as stuporous, excited, and malignant. Among them, malignant catatonia, which features fever and dysautonomia, is associated with increased morbidity and mortality [8,12]. A systemic review revealed that the prevalence of catatonia associated with medical conditions in hospitals and neurological department settings is approximately 13% to 21%, most commonly induced by inflammation of the CNS, followed by miscellaneous factors and neural injury. Additionally, SLE accounts for 8% of the catatonia cases caused by CNS inflammation [13]. Early diagnosis of catatonia in patients with SLE can be challenging due to the fact that it requires multidisciplinary evaluation. Moreover, this manifestation is not described in the 19 formal neuropsychiatric symptoms of NPSLE [14].

In consideration of the pathophysiology model for catatonia resulting from a dysfunctional restitutive dopamine system (mediated by GABAA) [12,15,16], this view may underlie the effectiveness of BZD GABAA agonism, NMDA-R antagonism, and ECT [6,9,11,12,15,16,17,18,19]. In 1983, lorazepam, a first-line treatment, was described as a successful treatment for catatonia [15,20]. A lorazepam challenge, moreover, represents a helpful diagnostic test for verifying catatonia in patients [7,18]. Many patients exhibit improvement with a single dose (2 mg) of intravenous lorazepam. Following the challenge, a maintenance dose of 2 mg lorazepam every 4 to 6 h is suggested [7].

### 2.2. TPE Is an Effective Treatment for CNS Lupus

In the treatment of CNS-associated SLE, TPE demonstrated improvement in the prognosis of 74% of cases in one previous study [21]. Boeke et al., reviewed 37 patients with SLE catatonia, discovering that of these patients, 86% lacked a previous psychiatric diagnosis, 81% underwent treatment with BZDs, and 27% underwent a combination treatment of TPE and high-dose steroids along with immunosuppressant therapy. Half of the patients subjected to the combination treatment rather than ECT showed favourable prognosis, indicating that TPE can be considered an effective treatment option for SLE catatonia [1]. Notably, however, ECT can pose certain ethical issues in adolescents and children [22]. ECT, glutamatergic agents, and atypical antipsychotics may be viable later-line catatonic therapy alternatives when BZDs are ineffective or contraindicated [6]. ECT works synergistically with BZDs and demonstrates response rates of up to 80% [9,17,23,24,25,26], particularly when applied at a frequency of 3 times per week for at least 6 sessions [9]. Case reports have indicated that glutamatergic agents are safe and well tolerated both as monotherapies or in combination with BZDs [6,27]. With the potential to worsen catatonia or cause progression to malignant catatonia, atypical antipsychotic agents are considered the last step in the treatment of catatonia [6,28,29]. Antipsychotics should be administered in combination with a BZD, and low-potency atypical agents are preferred. Typical high-potency antipsychotics should generally not be prescribed to patients with catatonia [9]. Supportive management is essential in cases of catatonia [6]. In addition to treatment of the catatonic symptoms, full resolution often requires treatment of the underlying disorder [11]. 

### 2.3. PP Is Potential an Effective Treatment for CNS Lupus Combined with Catatonia

In our patient, lorazepam therapy eventually commenced and resulted in effective benefits. However, the patient had another notable period of improved consciousness during hospitalisation, when PP improved her cognitive function. At this point, lorazepam treatment was only administered as needed for sedation during examination. Only a few case studies have reported the possible use of PP in intractable cerebral SLE [30,31]. The 2005 report that PP is an efficient treatment option for paediatric autoimmune neuropsychiatric disorders associated with streptococcal infections suggested that psychiatric symptoms related to immune dysfunction—such as SLE—could be improved by immune-modulatory treatment as well [30]. The gradual improvement following the first PP session in our case supports this assumption and the use of PP in cases of SLE associated with severe psychiatric symptoms. 

PP is a therapeutic option for lupus with CNS involvement [32,33,34]. However, only limited literature has demonstrated the positive benefits of PP in patients with SLE and catatonia [35]. A case report in 2012 [23] presented a 15-year-old patient with underlying SLE being diagnosed as having catatonia. She received PP for symptom management but with minimal improvement in catatonic symptoms, which were eventually controlled by ECT. To our knowledge, this is the only report describing a patient with catatonia and underlying SLE receiving PP, which demonstrated limited efficacy.

In conclusion, early diagnosis of catatonia is indispensable. Though the clinical consequences of PP treatment are not fully understood, PP followed by immunosuppressive therapy may be a therapeutic approach for patients with catatonia and underlying autoimmune disease, especially lupus with CNS involvement.

## Figures and Tables

**Figure 1 jcm-11-06670-f001:**
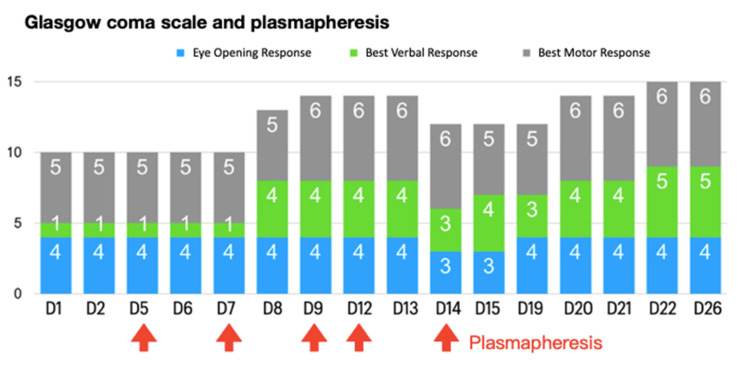
Glasgow coma scale and plasmapheresis during the course.

**Figure 2 jcm-11-06670-f002:**
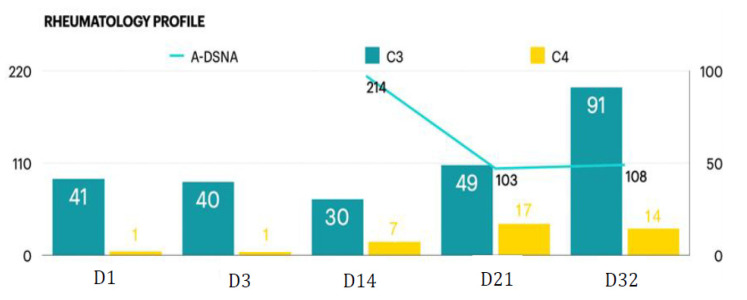
Rheumatology profile during the course.

**Table 1 jcm-11-06670-t001:** DSM-V diagnostic criteria for catatonia. Patient was diagnosed with catatonia following their matching with 4 clinical features accompanied by at least 3 of the aforementioned symptoms.

Feature	Description	Yes/No	Feature	Description	Yes/No
Stupor	Lack of psychomotor activity	Yes	Mannerism	Performing odd depictions of normal actions	No
Catalepsy	Maintenance of postures passively induced against gravity	No	Stereotypy	Repetitive purposeless stereotyped movements	No
Waxy flexibility	Initial rigidity prior to reduction in resistance to positioning by examiner	Yes	Agitation	Unpredictable and not influenced by external stimuli	No
Mutism	Very little to no verbal output	No	Grimacing	Odd or forced facial expression	No
Negativism	Lack of response to instructions of external stimuli	Yes	Echolalia	Mimicking others’ speech	No
Posturing	Spontaneous maintenance of postures	Yes	Echopraxia	Mimicking others’ movements	No

## Data Availability

Not applicable.

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
