# Peer review of "Plasmapheresis for a Patient with Catatonia and Systemic Lupus Erythematosus: A Case Report and Literature Review"

_jcm, 2022, doi:10.3390/jcm11226670_

Round 1

Reviewer 1 Report

This paper discusses an interesting case of SLE patient with CNS involvement characterized by catatonia and treated with plasma exchange.
Here are my comments:
- In the introductory part, at the time of diagnosis, please specify the antibody titer (ANA and anti-dsDNA), complement values, and proteinuria. Please specify how much glucocorticoids the patient received.
- The diagnosis of SLE was made according to what criteria? ACR 1997? Please specify it
- the patient subsequently receives PDN 40 mg daily. Has she also been prescribed hydroxychloroquine? If not, specify why.
- Is there any information regarding anti-p-protein antibodies and anti-neuron antibodies? Have anti-phospholipid antibodies been tested?
- Have infectious agents been ruled out? Has a cerebrospinal fluid analysis been performed? 
- It says PP in the text, but in the main test it is not clear what it refers to. It is specified only in the abstract. Please add it in the main test
- The patient was prescribed azathioprine. Specify why she did not take cyclophosphamide or mycophenolate mofetil since they are the two drugs of choice for the treatment of central nervous manifestations Moreover, it is unclear why PP was chosen since it is not recommended by the guidelines (EULAR 2019). Why the patient did this procedure and not cyclophosphamide? please specify the reasons for this choice
- In Figure 1, specify what the letters E, V, M stand for.
-in Figure 2 write "laboratory parameters" instead of rheumatology profile. Also unclear are the numbers on the x-axis. What does 11/17, 11/30, etc. mean?

Author Response

Dear reviewer,

  Thank you for providing these insights to our manuscript. The suggestions and the replies are listed below:

Reviewer

Suggestion

Reply

Revision

Reviewer 1

In the introductory part, at the time of diagnosis, please specify the antibody titer (ANA and anti-dsDNA), complement values, and proteinuria. Please specify how much glucocorticoids the patient received.

Thank you for your suggestion. We have detailed the relative data and reference inside the revised manuscript. The use of steroid therapy in outer hospital, however, is not available due to the patient did not afford the relative medical records and outpatient medication to us.

Owing to the progressive symptoms, she had visited a local hospital. In laboratory examinations at local hospital, positive findings for antinucleotide antibody (ANA) (1:320), anti-double-strand DNA (433 IU/mL. normal range <10 IU/mL), and low complement C3 (36 mg/dL. Ref. 90-180 mg/dL)/C4 (2.9 mg/dL. Ref. 10-40 mg/dL) levels were noted. According to medical records by local hospital, proteinuria, which is 3+ by urine stick examination, was noted as well.

Reviewer 1

The diagnosis of SLE was made according to what criteria? ACR 1997? Please specify it

The diagnosis of SLE is made according to SLICC criteria and EULAR/ACR criteria(2019) by available data in our hospital. However, in combination with the medical records in outer hospital, the patient is able to be diagnosed with SLE by EULAR/ACR criteria(2019) as well.

After the visit at local hospital with the diagnosis of systemic lupus erythematous(SLE), her parents took her to our rheumatology outpatient department (OPD) for a second opinion. A retrospective evaluation of medical history and laboratory data re-confirmed the diagnosis of SLE by EULAR/ACC criteria(2019)

Reviewer 1

the patient subsequently receives PDN 40 mg daily. Has she also been prescribed hydroxychloroquine? If not, specify why.

Due to severe disease with proteinuria, low C3/C4 and elevated anti-dsDNA by outer hospital, initial steroid therapy is reasonable with this patient before data rechecked. That is the reason why the patient receives PDN 40mg daily in the initial outpatient tracing.

Owing to the progressive symptoms, she had visited a local hospital. In laboratory examinations at local hospital, positive findings for antinucleotide antibody (ANA) (1:320), anti-double-strand DNA (433 IU/mL. normal range <10 IU/mL), and low complement C3 (36 mg/dL. Ref. 90-180 mg/dL)/C4 (2.9 mg/dL. Ref. 10-40 mg/dL) levels were noted. According to medical records by local hospital, proteinuria, which is 3+ by urine stick examination, was noted as well.

Reviewer 1

Is there any information regarding anti-p-protein antibodies and anti-neuron antibodies? Have anti-phospholipid antibodies been tested?

We don’t have further information with anti-p-protein and anti-neuron antibodies. The data of anti-phospholipid antibodies is presented in revised manuscript.

However, ANA (speckled; 1:1280), anti-dsDNA (362.7 IU/mL), and ribosome-P (178.45 units) were positive. Besides low values of complements in C3 (40 mg/dL) and C4 (1 mg/dL), and high anti-phospholipid IgG (29.75 GPL. Ref. <15 GPL) were revealed.

Reviewer 1

Have infectious agents been ruled out? Has a cerebrospinal fluid analysis been performed?

Infection, especially CNS infection, is the most important differential diagnosis in our patient. We have performed completely infectious surveys, including cerebrospinal fluid studies, all of which has ruled out the possibility with infection.

Laboratory examinations revealed mild normocytic anaemia with haemoglobin concentration and haematocrits levels of 9.7 g/dL and 28.8%, respectively. Creatinine level was 0.54 mg/dL. There were no evidence with active infection by urine, blood and cerebrospinal fluid studies.

…A cerebrospinal fluid study demonstrated negative findings for infection. Electroencephalography performed 3 times demonstrated negative findings. Consciousness returned to E4V4M6 (Glasgow coma scale score) after the fourth PP.

Reviewer 1

The patient was prescribed azathioprine. Specify why she did not take cyclophosphamide or mycophenolate mofetil since they are the two drugs of choice for the treatment of central nervous manifestations Moreover, it is unclear why PP was chosen since it is not recommended by the guidelines (EULAR 2019). Why the patient did this procedure and not cyclophosphamide? please specify the reasons for this choice

The treatment with mycophenolate mofetil and cyclophosphamide are not considered in this patient with acute and severe symptoms due to the payment by TW-NHI, the health insurance system in Taiwan. The choice of plasmapheresis is also depend on the payment criteria to SLE with CNS involvement by TWI-NHI criteria as well.

After admission to the rheumatology ward, a nephrologist was consulted for plasmapheresis (PP), which is supplied by Taiwan national health insurance(TW-NHI) system, with the diagnosis of lupus with central nerve system involvement.

Reviewer 2 Report

I read this case report of plasmapheresis (PP) use in patients with catatonia and lupus. Although authors have suggested they present a literature review of plasmapheresis use in lupus-induced catatonia, no PICO model is provided. I recommend the authors provide a PICO model to ascertain the total number of cases where PP was used as a treatment modality.

It appears that the patient seemed to have developed insomnia and depressive symptoms soon after the initiation of prednisolone 40mg a day. Could that be steroid-induced?

The case can be summarised. There seemed to be some repetitions, such as yelling and also, and a description of delusion symptoms may not be necessary. This will help expand the PP part.

It was good that the authors explained the lorazepam test to the readers. Not everyone may know it.

Can you please provide the full form for the table and figure abbreviations?

Author Response

Dear reviewer,

  Thank you for providing these insights to our manuscript. The suggestions and the replies are listed below:

Reviewer

Suggestion

Reply

Revision

Reviewer 2

It appears that the patient seemed to have developed insomnia and depressive symptoms soon after the initiation of prednisolone 40mg a day. Could that be steroid-induced?

The differential diagnosis with steroid induced psychotic symptoms in this patient. However, with severe symptoms with initial diagnosis of SLE with CNS involvement, steroid therapy is not suitable to be omitted during hospitalisation. With successful therapy with plasmapheresis and no further psychotic symptoms after discharge under steroid, it is believed that the psychotic symptoms in this patient are not induced by steroid therapy.

the patient’s consciousness recovered to normal (E4V5M6) within 1 week of commencing PP therapy (Figure1). In a stable condition, the patient was discharged on day 28 with outpatient tracing with oral steroid therapy. Further medication with cyclophosphamide is adjusted in outpatient clinic.

Reviewer 3 Report

Catatonia has not been included among the 19 neuropsychiatric manifestations of SLE in the classification criteria for neuropsychiatric SLE due to its low specificity. Additionally, based on the clinical and laboratory data presented, it is questionable whether this patient had SLE-associated catatonia. According to the SLE classification criteria,  a number of specific clinical and laboratory manifestations are required. This young woman didn’t have any history of clinical symptoms/signs included in the classification criteria such as joint, skin, renal, oral, hematological, or pulmonary manifestations. She only had positive ANA, anti-dsDNA (362.7 IU/mL), and ribosome-P (178.45 units) antibodies, although the cut-off point for anti-dsDNA and ribosome-P was not mentioned, so autoantibody positivity cannot be estimated. In the first laboratory examinations, the presence of low complement C3/C4 levels is reported but the exact levels are not mentioned. Also, proteinuria was noted but when the patient was hospitalized, normal urine tests are reported. Based on the above-mentioned clinical/laboratory features, the patient does not criteria for defining SLE.

Additionally, it is not clear whether the patient had finally improved by the use of oral lorazepam or by the plasmapheresis introduction. Even for well-diagnosed cases with SLE, there is no high-quality evidence data from the literature supporting the use of plasmapheresis (just very sporadic case reports), making its use in cases with no definite SLE, as in the presented case, really controversial.

Author Response

Dear reviewer,

  Thank you for providing these insights to our manuscript. The suggestions and the replies are listed below:

Reviewer

Suggestion

Reply

Revision

Reviewer 3

The diagnosis of SLE

Thank you for your suggestion. The initial diagnosis with SLE is completely surveyed at outer hospital, which data is not fully available. However, According to SLICC criteria and EULAR/ACR criteria(2019), the patient is compatible with the diagnosis with SLE. The improvement with proteinuria may be resulting from the partial treatment by steroid therapy at outpatient clinic.

 fter the visit at local hospital with the diagnosis of systemic lupus erythematous(SLE), her parents took her to our rheumatology outpatient department (OPD) for a second opinion. A retrospective evaluation of medical history and laboratory data re-confirmed the diagnosis of SLE by EULAR/ACC criteria(2019)

Reviewer 3

efficacy with lorazepam and plasmapheresis therapy.

Catatonia, indeed, is not among the 19 neuropsychiatric manifestations of SLE. However, both contained overlapped psychotic symptoms, which made the patient is initially diagnosed with NPSLE and later diagnosed with catatonia by lorazepam examination. Lorazepam therapy, in previous experiences, can only relieved symptoms in catatonia. In our patient, she can only short-term recover normal condition after lorazepam injection, and back to psychotic condition hours later. Symptoms get fully subsided after times of plasmapheresis programme. The patient is in normal appearance without lorazepam therapy after discharge.

The use of plasmapheresis is initially introduced to this patient due to NPSLE. However, after reviewing exist evidences, there are weak evidence between plasmapheresis and catatonia. We introduce a successful case whose catatonia is relieved by the treatment.

Consciousness returned to E4V4M6 (Glasgow coma scale score) after the fourth PP.

Due to CNS lupus complicated by catatonia, PP was recommended by the rheumatologist. Regular oral lorazepam (2 mg, 4 times/day) was recommended for symptom control. Impressively, the patient’s consciousness recovered to normal (E4V5M6) within 1 week of commencing PP therapy (Figure1). Lorazepam therapy is omitted with improving symptoms. There are not recurrent psychotic symptoms. In a stable condition, the patient was discharged on day 28 with outpatient tracing with oral steroid therapy. Further medication with cyclophosphamide is adjusted in outpatient clinic.

Reviewer 4 Report

In this manuscript, the authors have described a case of a young female with NPSLE with catatonia symptoms who improved with the immunosuppressant treatment plus therapeutic plasma exchange.

My major comment concerns the use of the term plasmapheresis and therapeutic plasma exchange throughout the text.

Therapeutic plasma exchange is a technique of clearance of extracorporeal blood, by which plasma is removed. A variable volume of plasma is removed from the patient and replaced by replacement solutions that maintain oncotic volume and pressureThe term 'plasmapheresis' should be reserved for situations where only plasma removal is carried out without replenishment, as is the case with plasma donation by apheresis for transfusion use or subsequent industrial fractionation of plasma. This procedure extracts less plasma (around 600 ml), without replacement solution, in less time and with simpler separation techniques than those used in TPE.

Since they are two different procedures, I think it is very important for the readers to describe in detail which one they used on this patient.

Author Response

Dear reviewer,

  Thank you for providing these insights to our manuscript. The suggestions and the replies are listed below:

Reviewer

Suggestion

Reply

Revision

Reviewer 4

My major comment concerns the use of the term plasmapheresis and therapeutic plasma exchange throughout the text.

Since they are two different procedures, I think it is very important for the readers to describe in detail which one they used on this patient.

Thank you for your suggestion. The treatment with plasmapheresis is used on our patient at the moment. We have detailed the information in our revised manuscript.

After admission to the rheumatology ward, a nephrologist was consulted for plasmapheresis (PP), which is supplied by Taiwan national health insurance(TW-NHI) system, with the diagnosis of lupus with central nerve system involvement.

Round 2

Reviewer 1 Report

The authors addressed to all the comments made. The work is now ready for publication 

Author Response

Thank you for your affirmation.

Reviewer 3 Report

The authors have satisfactorily addressed my comments

Author Response

Thank you for your positive reply, much appreciated.